# Long noncoding RNA HOTAIR interacts with Y-Box Protein-1 (YBX1) to regulate cell proliferation

Siting Li[1,2,4,*], Qian Xiong[1,2,4,*], Minghai Chen[3,4], Bing Wang[1,2,4], Xue Yang[1,2,4], Mingkun Yang[1,2,4], Qiang Wang[5], Zongqiang Cui[3,4], Feng Ge[1,2,4]

**HOTAIR is a long noncoding RNA (lncRNA) which serves as an important factor regulating diverse processes linked with cancer development. Here, we used comprehensive identification of RNA-binding proteins by mass spectrometry (ChIRP-MS) to explore the HOTAIR-protein interactome. We were able to identify 348 proteins interacting with HOTAIR, allowing us to establish a heavily interconnected HOTAIR-protein interaction network. We further developed a novel near-infrared fluorescent protein (iRFP)-trimolecular fluorescence complementation (TriFC) system to assess the interaction between HOTAIR and its interacting proteins. Then, we determined that HOTAIR specifically binds to YBX1, promotes YBX1 nuclear translocation, and stimulates the PI3K/Akt and ERK/RSK signaling pathways. We further demonstrated that HOTAIR exerts its effects on cell proliferation, at least in part, through the regulation of two YBX1 downstream targets phosphoenolpyruvate carboxykinase 2 (PCK2) and platelet derived growth factor receptor β. Our findings revealed a novel mechanism, whereby an lncRNA is able to regulate cell proliferation via altering intracellular protein localization. Moreover, the imaging tools developed herein have excellent potential for future in vivo imaging of lncRNA–protein interaction.**

## Introduction

Long noncoding RNAs (lncRNAs) are RNA molecules that, despite being longer than 200 nts, lack the ability to encode for protein (Fang & Fullwood, 2016). Although being initially ignored as they were believed to be of no consequence, more recent work has found that lncRNAs play essential roles as regulators of a wide range of cellular processes (Quinn & Chang, 2016). Importantly, many lncRNAs have also been found to play key roles in specific diseases, including in many types of cancer (Lorenzen & Thum, 2016;

Kopp & Mendell, 2018). The primary means by which lncRNAs mediate biological effects is through interacting with specific proteins (Wan & Chang, 2010; Zhang et al, 2014). Hence, the identification of cancer-associated lncRNAs and their interacting partners represents an important and ongoing area of investigation.

HOX Transcript Antisense RNA (HOTAIR) is a 2,148-nt lncRNA that has been associated with cancer and which is encoded within the Homeobox C (HOXC) gene cluster of chromosome 12 (Tsai et al, 2010; Chu et al, 2011). HOTAIR regulates the proliferation and metastasis of cancer cells, and for many tumor types, it has been found to be predictive of patient prognosis and progression (Wan & Chang, 2010; Zhang et al, 2014). Via interacting with polycomb repressive complex 2 (PRC2) and regulating the targets downstream of this protein, HOTAIR can mediate cellular metastasis (Gupta et al, 2010; Wan & Chang, 2010; Li et al, 2013), serving as a molecular scaffold which links histone methylase and demethylase activity, resulting in a wide range of unique histone modifications to nearby chromatin (Tsai et al, 2010).

Several studies have identified unique roles for HOTAIR that are attributable to its protein interacting partners (Wu et al, 2013; Aiello et al, 2016; Xue et al, 2018). Although some HOTAIR target proteins are known, the fact that this lncRNA is associated with so many different cellular functions suggests that there may be additional targets which are currently not understood. Therefore, an effort to accurately and comprehensively explore the HOTAIR protein interaction network is critical to facilitate a complete understanding of the role of this lncRNA in health and disease.

Comprehensive identification of RNA-binding proteins by mass spectrometry (ChIRP-MS) is a technique for studying endogenous ribonucleoprotein complexes, which is robust across a wide range of expression level, from abundant housekeeping RNAs to relatively lowly expressed RNAs (Chu et al, 2015). In the present study, using ChIRP-MS method, we were able to identify 348 proteins interacting with HOTAIR, allowing us to establish a heavily interconnected HOTAIR-protein interaction network. We additionally developed and optimized a novel Trimolecular Fluorescence Complementation

[1]State Key Laboratory of Freshwater Ecology and Biotechnology, Institute of Hydrobiology, Chinese Academy of Sciences, Wuhan, China    [2]Key Laboratory of Algal Biology, Institute of Hydrobiology, Chinese Academy of Sciences, Wuhan, China    [3]State Key Laboratory of Virology, Wuhan Institute of Virology, Chinese Academy of Sciences, Wuhan, China    [4]University of Chinese Academy of Sciences, Beijing, China    [5]State Key Laboratory of Crop Stress Adaptation and Improvement, School of Life Sciences, Henan University, Kaifeng, China

Correspondence: gefeng@ihb.ac.cn; czq@wh.iov.cn
*Siting Li and Qian Xiong contributed equally to this work

(TriFC) system based on near-infrared fluorescent protein (iRFP), which we successfully used to assess interactions between HOTAIR and its newly identified binding partner Y-box protein-1 (YBX1). Through additional functional analyses, we found that the interaction of HOTAIR with YBX1 is at least partially responsible for mediating the effect of HOTAIR on cell proliferation. We were thus able to uncover a novel oncogenic role for HOTAIR consisting in the alteration of the intracellular localization of YBX1.

# Results

### Systematic identification of HOTAIR-interacting proteins

We first used qRT-PCR to measure the relative expression of HOTAIR in 11 different cell lines and found that HeLa cell line has abundant HOTAIR expression (Fig S1A). Our previous study showed that CRISPR is not suited for mediating HOTAIR deletion (Wu et al, 2018), so we used transient siRNA-mediated and stable shRNA-mediated methods to knockdown HOTAIR expression (Fig S1B). HOTAIR knockdown resulted in significantly inhibited cellular proliferation as determined via CCK-8 and colony formation assays (Fig S1C and D). When nude mice were implanted with siHO and shHO knockdown cells, the resultant tumors were smaller in terms of both mass and volume (Fig S1E–H) than those derived from control cells (siNC and shNC), suggesting reduced tumorigenicity of HeLa cells. HOTAIR knockdown HeLa cells also showed reduced migration and invasion (Fig S1I and J), as well as increased apoptotic cell death (Fig S1K). These results indicate that HOTAIR can mediate the functional regulation of cellular processes related to cancer progression, which is consistent with the results of previous research (Zheng et al, 2015).

To gain a comprehensive understanding of the biological role of HOTAIR and the molecular mechanisms underlying these roles, we next conducted a ChIRP-MS experiment (Fig 1A). We used RNase A–treated samples, as well as nontargeting probe groups as negative controls (Kaida et al, 2010; Berg et al, 2012). According to Chu and Chang (2018), ChIRPing RNase-treated chromatin in parallel is the best way to control for nonspecific background, which are preferred to nontargeting probes because some background noise may be probe-specific. In the present study, about 80% HOTAIR RNAs in the cells were pulled down by performing ChIRP, whereas only 0.25% of GAPDH DNA was retrieved, demonstrating that ChIRP-MS enriches HOTAIR RNA with high yield and specificity (Fig S2). In the RNase-treated group, neither HOTAIR nor GAPDH was retrieved (Fig S2). Thus, using RNase A treatment groups as controls, we next performed a SAINT (Significance Analysis of INTeractome) analysis (Choi et al, 2011) with the SAINTexpress software (Teo et al, 2014) to probabilistically scoring the identified HOTAIR-protein interactions. With a high degree of confidence based on a Bayesian false discovery rate of ≤0.05, we were ultimately able to screen 348 proteins that interact with HOTAIR (Table S1). We then compared the 348 proteins to the identification results of the nontargeting probe control group (Table S1), and found only one overlapping protein (Q6QXN6) in one of the three replicates of nontargeting probe control group, further confirming the specificity of the ChIRP-MS results.

We next conducted a Gene Ontology (GO) enrichment analysis and found that the GO molecular function which was most over-represented among these 348 proteins was "binding" (Fig 1B), suggesting that HOTAIR may play a multifunctional role in regulating cell biology via indirectly and directly binding with many targets. These HOTAIR-interacting proteins were linked to a wide range of cellular processes, indicating the essential regulatory role of HOTAIR (Fig 1B and Table S2).

We next used the STRING database to create a protein–protein interaction (PPI) network for HOTAIR-interacting proteins, revealing that 217 of our identified 348 proteins were linked in a PPI network (Fig 1C and Table S3). The 217 proteins were linked with multiple functional roles including "carcinogenesis," "cell proliferation," "cell apoptosis," "translation," and "translation" indicating that these proteins play a key role in the dynamic HOTAIR complex (Fig 1C and Table S4).

### Validation of novel HOTAIR-interacting proteins

Of these 348 identified HOTAIR-interacting proteins, we selected YBX1, cyclin-dependent kinase 6 (CDK6) and translationally controlled tumor protein (TCTP), which are pivotal in cancer progression for additional validation. After ChIRP, we performed Western blotting and detected all three proteins from experimental groups, whereas not from negative control groups (Fig 2A). We further confirmed the interaction of YBX1/CDK6/TCTP and HOTAIR by performing RNA immunoprecipitation (RIP) assay (Fig 2A). The above results show the reliability of our ChIRP-MS experiment.

### A novel iRFP-TriFC system for investigating lncRNA-protein interactions

TriFC is a simple and powerful tool ideal for visualizing the interactions between RNA and proteins in living cells (Hiatt et al, 2008; Shyu & Hu, 2008). However, at present there is no effective TriFC system for assessing lncRNA-protein interactions in living cells under physiological temperature. Here, we conducted a systematic screening of split reporters from a bacterial phytochrome-based iRFP. We selected two split sites between amino acids 97–98 and 123–124, which have high complementary efficiency in the iRFP-BiFC system (Chen et al, 2015), as polypeptide split sites for testing the potential for TriFC construction. The general principle of the iRFP-TriFC system is shown in Fig 2B. A schematic diagram of the plasmid constructs used for the TriFC system in this study was shown in Fig S3A. MS2 coat protein (MCP) protein was fused to the iRC fragment and tethered to its RNA operator *ms2* by an MCP–*ms2* interaction. The iRN fragment was fused to an RNA-binding protein candidate. The *ms2* sequence and the RNA sequence of interest were linked in the backbone plasmid pECFP-C1. The iRFP-TriFC system (including plasmids piRN, piRC-MCP, and pECFP-C1-ms2) was constructed based on this design. If the candidate RNA-binding protein interacted with the RNA sequence of interest, the association of the two iRFP fragments would produce a red TriFC signal. In this system, the two fragments of iRFP which do not fluoresce on their own will come together when the RNA and protein of interest interact, resulting in fluorescence and allowing for real-time imaging of these interactions. We first validated this system using the human polypyrimidine-tract-binding

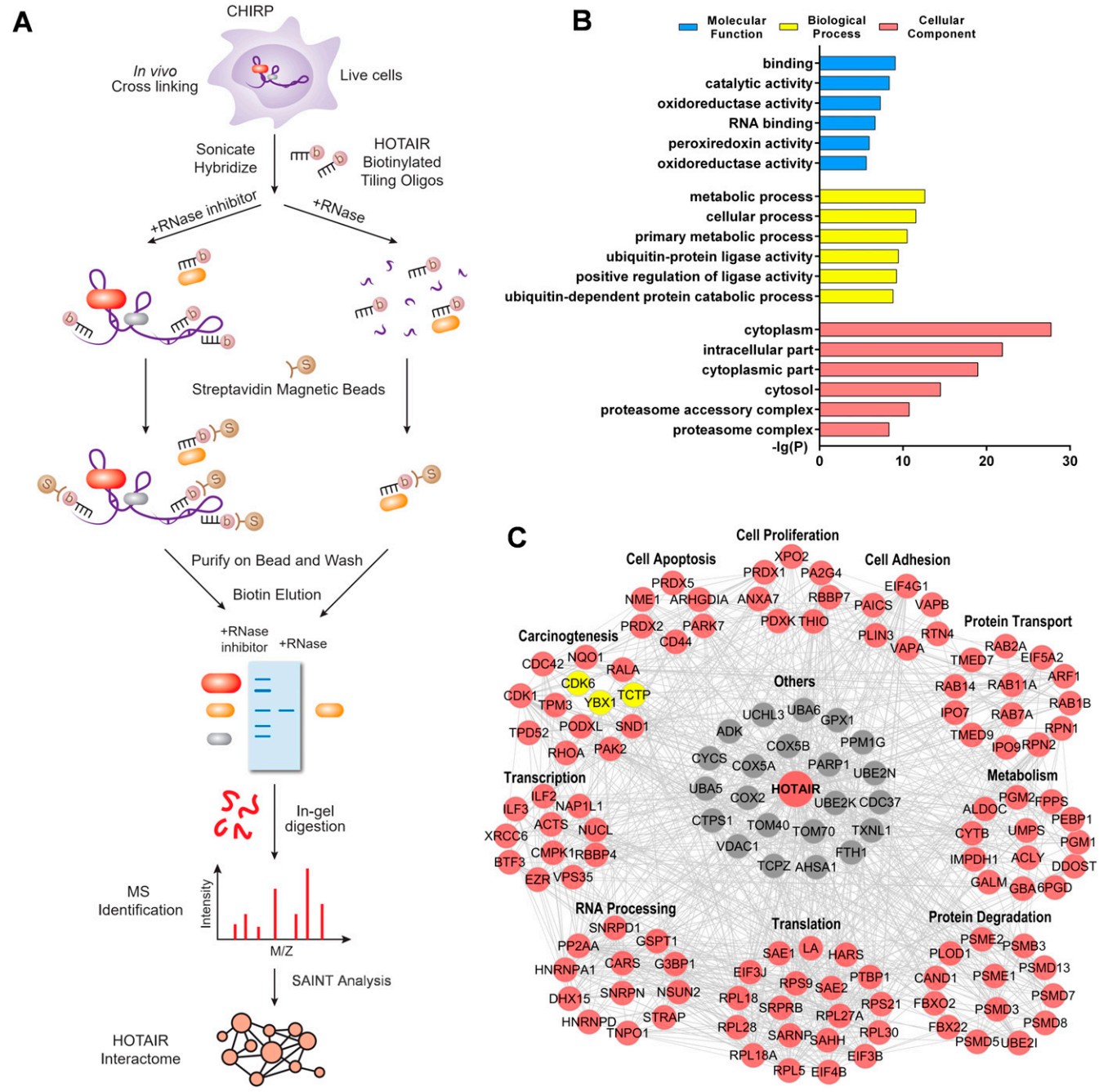

**Figure 1. Systematic Identification of HOTAIR-interacting Proteins.**
**(A)** Experimental overview. **(B)** A gene ontology enrichment analysis of the identified HOTAIR-interacting proteins, with the top six gene ontology terms being shown. **(C)** A protein–protein interaction network analysis of HOTAIR-interacting proteins, which were grouped based on their biological functions. Nodes with edges number in STRING ≥ 6 were shown. Yellow nodes indicate the proteins selected for further validation.

protein (PTB) protein and the CRSs of HIV-1 envelope (envCRS) mRNA pair (Black et al, 1996), which are well known to strongly interact and thus ideal for a proof-of-concept iRFP-based TriFC approach. We cloned the PTB coding gene and the envCRS mRNA into TriFC plasmids (Fig S3A), and these were then co-transfected into 293T cells. Results showed that the iRN97/iRC98 reporter pair did not emit any detectable fluorescence, whereas the iRN123/iRC124 reporter pair

emit weak but detectable complementary fluorescence (Fig S3B). Thus, the iRN123/iRC124 reporter pair was used for iRFP-TriFC system construction.

To optimize the iRFP-TriFC system and increase the complementary fluorescent signal, the iRC124 fragment was subjected mutagenesis at three sites (F164Y/D231Y/W308R; termed iRC124m) in the GAF (Pfam entry: PF01590) domain (Wahlestedt, 2013). After

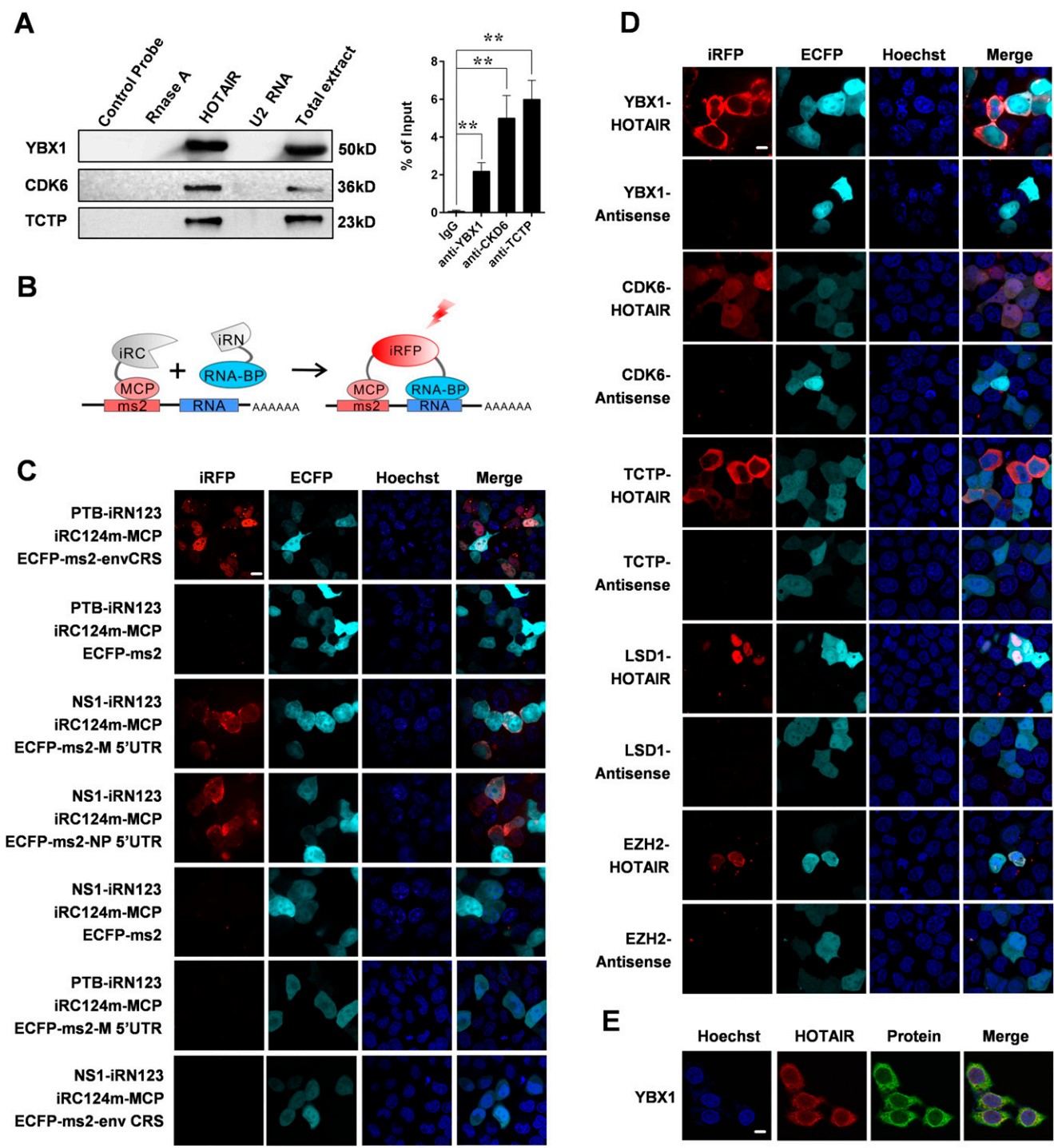

**Figure 2. A Novel iRFP-TriFC System for Investigating long noncoding RNA-protein interactions.**
**(A)** In vitro validation of interactions between HOTAIR and the indicated proteins. Left: ChIRP experiment was performed, followed by Western blotting. A nontargeting probe group (control probe), an RNase A–treated group and an unrelated RNA U2 group were used as negative controls. Right: RIP was performed to validate the RNA-protein interactions. Data are means ± SD of triplicate experiments. *P < 0.05 and **P < 0.01 (t test). **(B)** Schematic overview of TriFC constructs. iRN and iRC (gray segment) represent the iRFP N-terminal and C-terminal fragments, respectively. **(C)** Validation the iRFP-TriFC system with three known protein–RNA interaction pairs: PTB/envCRS; NS1/M 5'UTR; NS1/NP 5'UTR. ECFP-ms2 was used instead of ECFP-ms2-envCRS, ECFP-ms2-M 5'UTR, or ECFP-ms2-NP 5'UTR in the iRFP-TriFC system as a negative control. The PTB/M 5' UTR and NS1/envCRS protein–RNA pairs were also used as negative controls. **(D)** In vivo confirmation of protein interactions with HOTAIR using the iRFP-TriFC approach. **(E)** Co-localization of HOTAIR (red) and YBX1 (green) within HeLa cells as determined by immunofluorescence (IF) and RNA-FISH. Hoechst 33342 was used for nuclear staining, and a 63×, 1.4 NA, oil immersion objective lens was used for all fluorescent image capture. Scale bars: 10 μm.

transfection into HEK 293T cells, the EGFP-iRN123 and iRC124m-EGFP combination yielded threefold stronger complementary fluorescence than the EGFP-iRN123 and iRC124-EGFP combination did (Fig S3C). We therefore replaced iRC124 with iRC124m in the iRFP-TriFC system.

We next verified the optimized iRFP-TriFC system using three well-known protein-RNA interaction pairs: the PTB/envCRS pair used above, as well as the paring of the influenza A virus NS1 protein with the 5′ UTR of NP or M mRNAs (Park & Katze, 1995). All three pairs yielded strong complementary fluorescent signal as compared to negative controls (Fig 2C). Then we switched the proteins in the two RNA-protein interaction pairs above, using PTB/M 5′ UTR and NS1/envCRS protein-RNA pairs as two additional negative controls (Fig 2C). A quantitative analysis of the iRFP/ECFP fluorescence intensity ratio confirmed that these TriFC signals were significantly greater than those produced by negative control constructs (Fig S3D). This TriFC-reconstituted iRFP also exhibited photophysical properties comparable to those of unmodified iRFP (Fig S3E). The expression of all the recombinant proteins expressed in the iRFP-TriFC system was examined by Western blotting (Fig S3F). As these split iRFP reporters are stable at 37° C, the low temperature pre-incubation step required by previous TriFC systems can be omitted (Yin et al, 2013; Han et al, 2014). This novel iRFP-TriFC system thus offers an effective means of imaging ncRNA–protein interactions in living cells under physiological temperature.

We next used this iRFP-TriFC system to assess the interaction between HOTAIR and YBX1/CDK6/TCTP in living cells. Antisense (an antisense-HOTAIR fragment) RNA was used as negative control for HOTAIR. We were able to detect complementary fluorescence in the whole cells when co-expressing HOTAIR and CDK6, whereas cells co-expressing HOTAIR and YBX1/TCTP exhibited bright cytoplasmic fluorescence (Fig 2D). The expression of the recombinant YBX1-iRN123, CDK6-iRN123, and TCTP-iRN123 was examined by Western blotting (Fig S3F). For comparison and verification, we used the iRFP-TriFC system to examine the interaction between HOTAIR and its two well-known interacting partners, lysine demethylase 1A (LSD1) and enhancer of zeste 2 PRC2 subunit (EZH2) (Gupta et al, 2010; Tsai et al, 2010). Results showed that the interaction between HOTAIR and LSD1, EZH2 occurred in cell nucleus (Fig 2D). In further, co-localization experiment of HOTAIR and YBX1 also showed that the interaction of HOTAIR and YBX1 occurred in cytoplasm (Fig 2E). These results suggested that the iRFP-TriFC system can both allow for real time assessment of lncRNA–protein interactions, and can facilitate the investigation of the subcellular localization of these interactions.

## The YBX1 cold shock domain (CSD) specifically interacts with CACC motifs in HOTAIR

Then we selected proteins for further validation and functional study. We used three criteria for screening: First, the fold change given by SAINTexpress is above 50. Second, the protein is a reported RNA-binding protein, referring to two global RNA-binding protein profiling performed in HeLa cells (Castello et al, 2012; Trendel et al, 2019), in other words, this protein should be in at least one of the two RNA-binding protein lists. Third, the protein is in the PPI network generated by STRING database (Table S3). Thirty-one proteins remain after screening (Table S1, protein name marked

in red), in which YBX1 has been reported to interact with several oncogenic lncRNAs, including TP53TG1 (Diaz-Lagares et al, 2016), HULC (Li et al, 2017), HOXC-AS3 (Zhang et al, 2018), and lincNMR (Castello et al, 2012), thereby regulating cancer progression and oncogene expression. Considering its pivotal role in RNA regulated cancer biology (Lyabin et al, 2014), YBX1 was selected for further functional studies. YBX1 is a 36 kD protein composed of an N-terminal alanine/proline-rich domain (A/PD), a C-terminal domain (CTD), as well as a CSD (Fig 3A) (Lyabin et al, 2014). The CSD region is similar to cold shock proteins found in bacteria with respect to its tertiary structure (Kloks et al, 2002), whereas the A/PD and CTD regions are thought to lack any intrinsic structure (Kloks et al, 2002; Gunasekaran et al, 2016). The TriFC results showed that all three YBX1 domains could interact with HOTAIR (Fig 3B). However, only the CSD–HOTAIR interaction exhibited the same cellular localization as YBX1 and HOTAIR (Fig 3B). This is not surprising because the CSD domain of YBX1 has been reported to be vital for RNA-binding (Kloks et al, 2002), whereas CTD and A/PD domains may serve to stabilize these interactions (Eliseeva et al, 2011). The expression of the recombinant A/PD-iRN123, CSD-iRN123, and CTD-iRN123 was verified by Western blotting (Fig S3F).

HOTAIR is known to be highly structured, possessing four independent domains (Fig 3C) (Somarowthu et al, 2015). Using our iRFP-TriFC system, we were able to determine that three HOTAIR domains (HD1, HD2, and HD4) can interact with YBX1 (Fig 3D and E). CACC and CAUC motifs were reported to be specific high-affinity RNA-binding motifs for YBX1, and the second adenosine (A2) and the fourth cytidine (C4) in the CACC/CAUC motif are important for YBX1 recognition (Eliseeva et al, 2011). HOTAIR harbors 13 CACC motifs and three CAUC motifs (Fig 3C). We mutated all the CACC motifs in HOTAIR to CUCG (named HOTAIRm), whereas all CAUC motifs in HOTAIR to CUUG (named HOTAIRm′). The results showed that YBX1 could interact with HOTAIRm′, but not HOTAIRm. We also showed that YBX1 failed to interact with CACC mutated HOTAIR domains (HD1m, HD2m, and HD4m) (Fig 3D and E). So we can conclude that the CACC motifs in HOTAIR are indispensable for HOTAIR–YBX1 interaction.

Then we sought to explore the specificity of interaction between YBX1 domains and RNA. TriFC system was used to examine the interaction of three YBX1 domains with G3BP1 mRNA, a determined YBX1-interacting mRNA harboring nine CACC motifs (Somasekharan et al, 2015), as well as Antisense and HOTAIRm, which were already proved to have no interaction with full-length YBX1 (Figs 2D and 3D). As expected, CSD domain can interact with G3BP1 mRNA, and the cellular localization of CSD-G3BP1 mRNA is the same as full-length YBX1 and G3BP1 mRNA (Fig S4A). It is intriguing that as no interaction was observed between CTD domain and any of the three RNAs, whereas the A/PD domain interacts with all three RNAs (Fig S4A–C). More interestingly, the interaction between A/PD and the three RNAs is ubiquitous in cells, which is the same as HOTAIR and A/PD. Thus, we deduce that the specific binding of YBX1 and RNA depends on CSD domain.

## HOTAIR promotes the Ser102 phosphorylation and nuclear translocation of YBX1

To explore the physiological importance of the interaction between YBX1 and HOTAIR, we first knockdown or overexpress HOTAIR and determined that the expression change of HOTAIR did not alter the

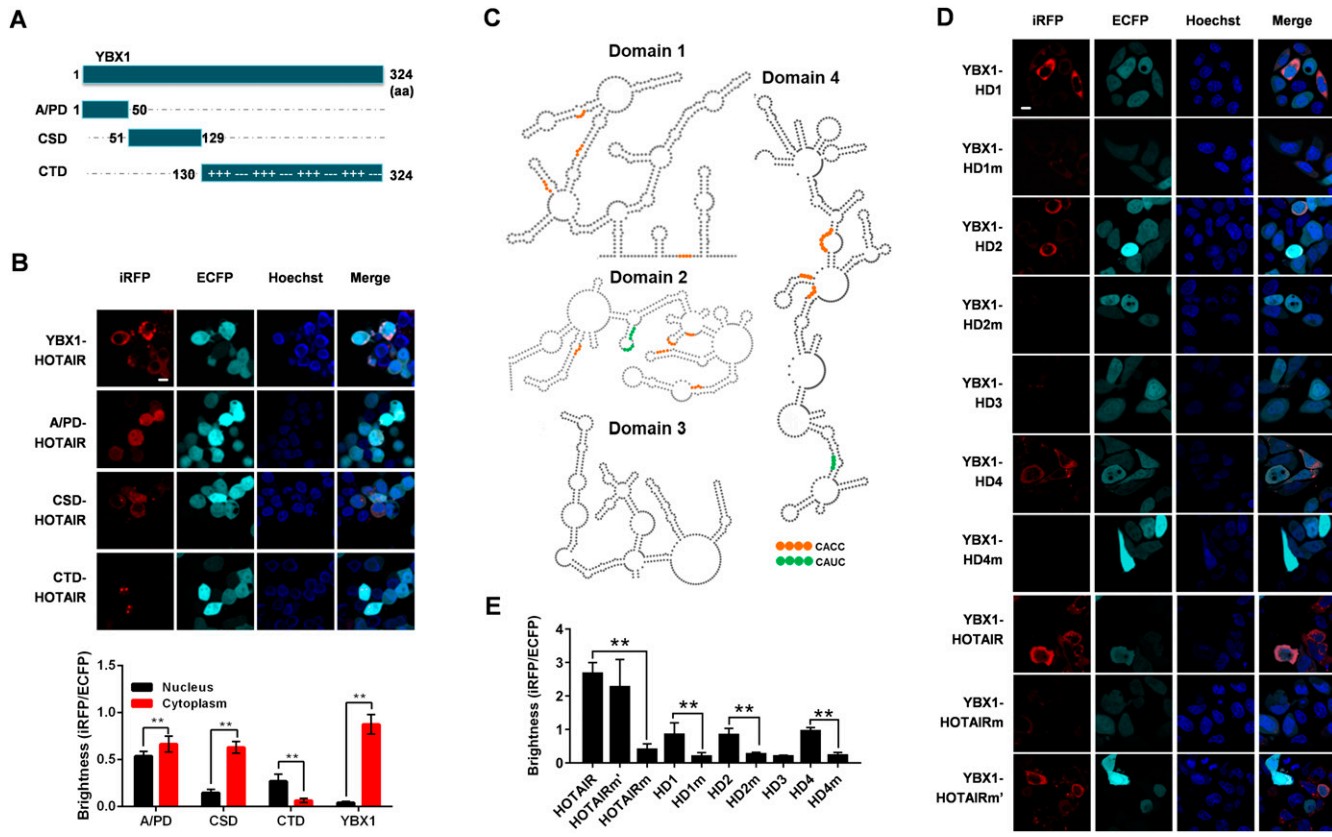

**Figure 3. The cold shock domain of YBX1 specifically interacts with HOTAIR CACC motifs.**
**(A)** Schematic of YBX1 protein domains. **(B)** Up: The interaction between HOTAIR and truncated or full-length YBX1 was visualized by iRFP-TriFC system in live cells. Down: Quantification of the fluorescence (iRFP/ECFP) for different combinations of TriFC interacting pairs. **(C)** Schematic of HOTAIR domains, with the CACC/CAUC YBX1-binding motifs indicated by orange/green. The image of HOTAIR domains is based on the HOTAIR secondary structure published by Srinivas et al (Somarowthu et al, 2015). **(D)** Use of the iRFP-TriFC system for In vivo imaging of interactions between YBX1 and HOTAIR, HOTAIRm (CACC mutated to CUCG), HOTAIRm' (CAUC mutated to CUUG), or the different domains of HOTAIR (HD1-HD4), or HOTAIRm (HD1m, HD2m, or HD4m). **(D, E)** Quantification of the fluorescence (iRFP/ECFP) for different interacting pairs shown in (D). **(B, D)** In both (B) and (D), Hoechst 33342 was used for nuclear staining. Scale bars: 10 $\mu$m. **(B, E)** In (B, E), data are means ± SD (n = 60) of triplicate experiments. *$P$ < 0.05 and **$P$ < 0.01 ($t$ test).

mRNA or protein expression of YBX1 (Fig 4A, B, and D). YBX1 is known to be initially localized to the cytoplasm of cells, only undergoing nuclear translocation in response to certain intra- or extra-cellular signals (Eliseeva et al, 2011). So we next assessed whether HOTAIR can affect YBX1 nuclear translocation. When we knocked down the expression of HOTAIR in HeLa cells, the YBX1 detected in nucleus (N-YBX1) was significantly reduced (Figs 4B and C and S5A and B). The same phenomenon was also found in other cancer cell lines (Fig S5C). To further verify the effect of HOTAIR on the N-YBX1 expression, we overexpressed HOTAIR as well as HOTAIRm with mutated YBX1-binding motifs in HeLa cells. Significantly increased expression of N-YBX1 proteins was detected after HOTAIR overexpression, whereas HOTAIRm overexpression has no effect on YBX1 subcellular locali-zation (Fig 4D and E). The quantification of YBX1 Western blotting signals showed significantly up-regulation of YBX1 in cytoplasm (C-YBX1) after HOTAIR knockdown, and significantly down-regulation of C-YBX1 after HOTAIR overexpression (Figs 4B and S5A). However, the immunofluorescent results only showed a slight tendency of up- or down-regulation after HOTAIR knockdown or overexpression (Figs 4C and E and S5B). This is because YBX1 is mainly located in the cytoplasm under normal cell condition (Eliseeva et al, 2011), which

was also indicated by the immunofluorescent results of the present study (Fig 2E). When we knockdown or overexpress HOTAIR, only a small portion of YBX1 will be prevented from or undergo nuclear translocation, which is not so evident compared with the high C-YBX1 expression. In summary, these results suggest that HOTAIR con-tributes to YBX1 nuclear translocation.

YBX1 nuclear translocation is reported to be associated with the phosphorylation of Ser102 (S102) within its CSD domain (Eliseeva et al, 2011). To verify this, we constructed mutants of YBX1 protein which mimicked either a constitutively phosphorylated (S102D) or non-phosphorylated (S102A) state. When overexpressed in HeLa cells, EGFP-YBX1 (wild-type) can be detected in both cytoplasm and nuclear. However, EGFP-YBX1 (S102A) were mainly detected in cy-toplasm and was almost absent in nucleus, whereas EGFP-YBX1 (S102D) was primarily found in the nucleus (Fig 4F). These results proved that S102 phosphorylation is crucial for YBX1 nuclear local-ization. We then tried to determine whether the phosphorylation of YBX1 S102 was affected by HOTAIR expression. We found that YBX1 S102 phosphorylation was substantially decreased after HOTAIR knockdown (Figs 4B and S5A), whereas significantly increased upon HOTAIR overexpression (Fig 4D). These results suggest that

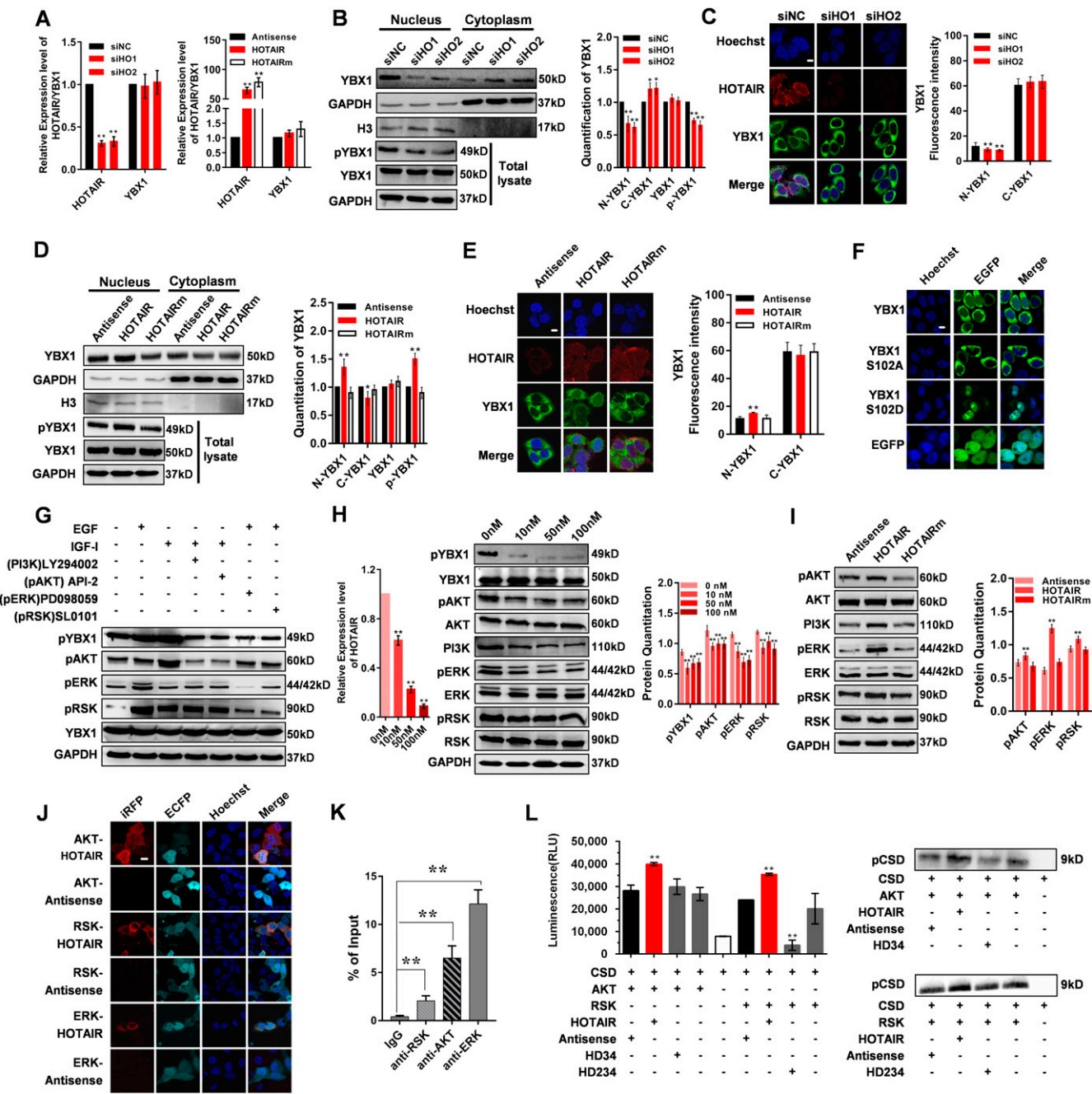

**Figure 4. HOTAIR regulates YBX1 nuclear translocation via pAKT and pRSK.**
**(A)** HOTAIR and YBX1 mRNA expression in HOTAIR knockdown or overexpression HeLa cells. Left: HOTAIR and YBX1 mRNA expression in HeLa cells transfected with siHO1/siHO2/siNC as measured by qRT-PCR. Right: HOTAIR and YBX1 mRNA expression in HeLa cells overexpressed with Antisense, HOTAIR, or HOTAIRm as measured by qRT-PCR. **(B)** Left: Western blotting analysis of cytoplasmic, nuclear, and total YBX1 and YBX1 p-S102 in HeLa cells transfected with siHO1/siHO2/siNC. GAPDH was used as a loading control for total and cytoplasmic protein, whereas H3 was used as a loading control for nuclear protein. Right: Quantification of the Western blot signals. **(C)** Left: Subcellular YBX1 protein (green) and HOTAIR (red) localization in HeLa cells transfected with siHO1/siHO2/siNC as determined by IF and RNA-FISH, respectively. Right: Quantification of nuclear and cytoplasmic YBX1 fluorescence (mean gray value). **(D)** Left: Western blotting analysis of cytoplasmic, nuclear, and total YBX1 and YBX1 p-S102 in HeLa cells overexpressing HOTAIR, HOTAIRm, or Antisense. GAPDH was used as a loading control for total and cytoplasmic protein, whereas H3 was used as a loading control for nuclear protein. Right: Quantification of the Western blot signals. **(E)** Left: Subcellular YBX1 protein (green) and HOTAIR (red) localization in cells overexpressing Antisense, HOTAIR, or HOTAIRm as determined by IF and RNA-FISH, respectively. Right: Quantification of nuclear and cytoplasmic YBX1 fluorescence. **(F)** Subcellular localization of EGFP-YBX1, EGFP-YBX1 S102A, and EGFP-YBX1 S102D in cells overexpressing YBX1/YBX1(S102A)/YBX1(S102D) as determined by confocal microscopy. **(G)** Western blotting quantitation of total pYBX1, pERK, pAKT, pRSK, and YBX1 levels in HeLa cells underwent serum starvation, followed by treatment with LY294002, API-2, or PD098059 and then EGF or insulin-like growth factor 1. **(H)** Left: HOTAIR was inhibited using siHO at a range of concentrations: 0, 10, 50, and 100 nM. The knockdown efficiency was determined by qRT-PCR. Middle: pYBX1, YBX1, pAKT, AKT, PI3K, pRSK, RSK pERK, and ERK levels in the HOTAIR knockdown HeLa cells as determined by Western blotting. Right: Quantification of the Western blot signals. **(I)** Left: pAKT, AKT, PI3K, pRSK, RSK pERK and ERK levels in HeLa cells overexpressing siHO, Antisense, HOTAIR, or HOTAIRm as determined by Western blotting. Right: Quantification of the Western blot signals. **(J)** Confirmation of HOTAIR and AKT/RSK/ERK protein interactions

HOTAIR can drive YBX1 nuclear translocation via promoting YBX1 S102 phosphorylation.

### HOTAIR regulates YBX1 phosphorylation via phosphorylated AKT kinase (pAKT) and phosphorylated RSK kinase (pRSK)

pAKT has been reported to phosphorylate YBX1 S102 (Sutherland et al, 2005). Phosphorylated ERK kinase (pERK) and pRSK are also known to contribute to YBX1 S102 phosphorylation (Astanehe et al, 2009; Imada et al, 2013; Donaubauer & Hunzicker-Dunn, 2016). To determine which of these pathways play a role in the context of HOTAIR expression, cells were treated using specific inhibitors of the phosphoinositide 3-kinase (PI3K)-Akt pathway (PI3K: LY294002 and AKT: API-2) or ERK/RSK pathways (ERK: PD098059 and RSK: SL0101), along with insulin-like growth factor 1 (IGF-I) or EGF, respectively. We detected significant activation of AKT/ERK/RSK in response to IGF-I/EGF, as determined by phosphorylated AKT/ERK/RSK (Fig 4G). All the above four inhibitors disrupted YBX1 phosphorylation (Fig 4G), suggesting that HOTAIR-mediated YBX1 phosphorylation can be achieved via both PI3K/Akt and ERK/RSK pathways. ERK is a proline-directed kinase that preferentially catalyzes the phosphorylation of substrates containing a Pro-Xxx-Ser/Thr–Pro sequence (Roskoski, 2012). However, YBX1 S102 lacks a +1 Pro, which makes it unlikely to be directly phosphorylated by ERK. RSKs are downstream effectors of the Ras-extracellular signal-regulated kinase (ERK)/MAPK signaling cascade (Roskoski, 2012), and activated ERKs can phosphorylate and activate RSKs (Dalby et al, 1998). Hence, it is probable that ERK regulates the phosphorylation of YBX1 S102 by activating its substrate, RSK.

Then, we tested the functional effect of HOTAIR on the PI3K/Akt and ERK/RSK pathways. Upon HOTAIR inhibition, we found that pAKT, pERK, pRSK, and PI3K were all significantly down-regulated (Figs 4H and S5D), whereas after HOTAIR overexpression, all four of them were up-regulated (Fig 4I). However, total AKT, RSK, and ERK expression was not affected by changes in HOTAIR expression (Fig 4H and I and S5D). We also confirmed that pYBX1, PI3K, pAKT, pERK, and pRSK were all down-regulated in HOTAIR-knockdown xenografts in comparison to control groups (Fig S5E), further supporting a role for HOTAIR in regulating PI3K/Akt and ERK/RSK–mediated YBX1 S102 phosphorylation.

Using our iRFP-TriFC approach, we were able to further establish that HOTAIR directly interacts with AKT, ERK, and RSK, which was also verified by RIP experiment (Fig 4J and K). Then we conducted YBX1 co-immunoprecipitation after overexpressing or knocking down HOTAIR. Results showed that pAKT/pERK/pRSK in the immunoprecipitates is increased after HOTAIR overexpression and reduced after HOTAIR knockdown, whereas the total immunoprecipitated AKT/ERK/RSK level was unchanged (Fig S5F). The expression of the recombinant AKT-iRN123, RSK-iRN123, and ERK-iRN123 was examined by Western blotting (Fig S5G). These results indicated that HOTAIR may regulate

the phosphorylation status of YBX1 by binding and affecting the phosphorylation of AKT and RSK.

Next, we studied the exact binding region of HOTAIR and different kinases. We constructed three AKT deletion mutants (AKT D1, D2, and D3) and three RSK deletion mutants (RSK D1, D2, and D3) based on their protein domain (https://www.ebi.ac.uk/services) (Fig S6A and B). ERK cannot be divided because it has only one kinase domain. The iRFP-TriFC results showed that AKT D2/D3 and RSK D2 are responsible for HOTAIR binding (Fig S6C and D). The kinase binding regions on HOTAIR were also investigated by using iRFP-TriFC system. We found that AKT interacts with HD1/HD2, and RSK interacts with HD1 (Fig S6C and D). However, only the full-length HOTAIR could bond to ERK (Fig S6D). The expression of the recombinant AKT D1-iRN123, AKT D2-iRN123, AKT D3-iRN123, RSK D1-iRN123, RSK D2-iRN123, and RSK D3-iRN123 was examined by Western blotting (Fig S6E). HOTAIR is well known for its function as a scaffold recruiting PRC2 and the LSD1/CoREST/REST complex (Gupta et al, 2010; Tsai et al, 2010). Therefore, we speculate that HOTAIR may act as a scaffold to draw AKT, ERK/RSK, and YBX1 together, thus promoting the phosphorylation of YBX1.

Then, we explored whether HOTAIR RNA could directly affect phosphorylation of YBX1 S102 by using an in vitro kinase assay. The YBX1 CSD domain was in vitro expressed and purified. ADP-Glo Kinase Assay and the following Western blotting results showed that both AKT and RSK could phosphorylate the S102 in purified CSD domain (Fig 4L). When HOTAIR was added to the reaction, the luminescence was significantly increased compared with the control groups (Fig 4L), indicating that HOTAIR RNA can directly promote the activity of AKT and RSK toward the CSD domain. These results thus imply that HOTAIR binding may alter accessibility of S102 in the CSD domain to AKT and RSK kinases.

To further establish the importance of HOTAIR-AKT and HOTAIR-RSK interactions for YBX1 S102 phosphorylation, we generated HOTAIR truncated mutants, HD34 (HOTAIR HD3-HD4) and HD234 (HOTAIR HD2-HD3-HD4), which are not able to bind to AKT and RSK kinase, respectively. iRFP-TriFC system was used to confirm that HD34 and HD234 did not interact with the AKT and RSK kinase, respectively (Fig S6C and D). The in vitro kinase assay also revealed that HD34 and HD234 had no significant effect on the activity of AKT or RSK towards the CSD domain (Fig 4L). These results indicated that the HD1 and HD2 domains are required for AKT-HOTAIR interaction, and HD1 domain is indispensable for RSK-HOTAIR interaction.

### HOTAIR and YBX1 co-regulate downstream gene transcription

Once it localizes to the nucleus, YBX1 serves as a transcription factor regulating many genes associated with oncogenic processes (Stein et al, 2001). Hence, we hypothesized that HOTAIR may mediate tumorigenesis via regulating YBX1 and its target downstream genes. To verify this possibility, we first collected HOTAIR-regulated genes annotated in different cancer types, including cervical cancer

using the iRFP-TriFC system. **(K)** Immunoprecipitation of AKT/ERK/RSK retrieved endogenous HOTAIR in HeLa cells. Immunoprecipitation of IgG was used as negative control. **(L)** Activity of AKT and RSK against the S102 in cold shock domain domain of YBX1 (pCSD). HOTAIR RNA was added to the reaction to test whether it could enhance the Kinase reaction, with Antisense, HD234 and HD34 used as negative control. RLU, relative luminescence units. The pCSD level was determined by Western blotting with pYBX1 antibody. **(C, E, F, J)** In (C, E, F, J), Hoechst 33342 was used for nuclear staining. Scale bar, 10 $\mu$m. **(B, D)** In (B) and (D), the loading amount of cytoplasmic protein is about 1/10 of that of nuclear protein because YBX1 is mainly expressed in the cytoplasm. All data are given as means ± SD. *$P$ < 0.05 and **$P$ < 0.01 ($t$ test).

cells (Zheng et al, 2015), hepatocellular carcinoma (Wu et al, 2018), pancreatic cancer cells (Kim et al, 2012), gastrointestinal stromal tumors (Niinuma et al, 2012), and small-cell lung cancer (Ono et al, 2014). Then we collected YBX1-regulated genes identified via ChIP sequencing and ChIP-on-chip (Finkbeiner et al, 2009; Astanehe et al, 2012). We further used the SABiosciences ChIP search tool (https://chip-atlas.org/target_genes) to explore additional genes which may be bound by YBX1. The overlap of HOTAIR-regulated genes and YBX1-regulated genes were collected as putative co-regulated genes (Table S5). KEGG pathway enrichment analysis of this HOTAIR-YBX1 co-regulated genes showed that "PI3K/Akt signaling pathway (hsa04151)" was significantly enriched (Fig 5A and Table S5). Then we selected three genes, platelet derived growth factor receptor β (*PDGFRB*), *PCK2*, and *BCL2* Associated Agonist of Cell Death (*BAD*), which are involved in this pathway for further study. When we knocked down HOTAIR or YBX1 in HeLa cells, all three genes were significantly decreased at both mRNA and protein level, confirming the regulatory roles of both HOTAIR and YBX1 on these genes (Fig 5B and C). Then we selected *PCK2* and *PDGFRB* for further studies, for their important roles in cell proliferation (Sachinidis et al, 2002; Montal et al, 2015). We found that the expression of *PCK2* and *PDGFRB* was significantly reduced in HOTAIR-knockdown xenografts (Figs 5D and S7), which is consistent with the results of in vitro studies (Fig 5B and C). When we inhibited the expression of either or both of the genes, significantly suppressed cell proliferation was observed (Fig 5E–G). In contrast, cell proliferation increased significantly when the two genes were overexpressed separately or simultaneously (Fig 5E–G). These results suggest that HOTAIR can regulate cell proliferation by regulating the expression of *PCK2* and *PDGFRB*. We then confirmed that the *PCK2* and *PDGFRB* promoter regions were bound by YBX1 through ChIP assay (Fig 5H). YBX1 has been reported to bind the inverted CCAAT box (Y-box) sequence in target gene promoters (Ise et al, 1999). Multiple inverted Y-box sequences were found within the promoters of both *PCK2* and *PDGFRB* (genome browser: uc001wlt.4 and uc003lro.4), which may be bound and activated by YBX1 (Fig 5I). Using a luciferase reporter gene assay, we further confirmed the ability of YBX1 to bind the *PCK2* and *PDGFRB* promoters (Fig 5J). Thus, we deduce that YBX1 may play an activating role on the transcription of *PCK2* and *PDGFRB* by binding to their promoter region. Taking together, our results suggest that HOTAIR is able to regulate the proliferation of cancer cells by modulating YBX1 nuclear localization, promoting in turn *PCK2* and *PDGFRB* expression. Based on the above findings, we propose a model wherein HOTAIR effectively serves to drive the expression of downstream genes such as *PCK2* and *PDGFRB* via the regulation of YBX1 nuclear translocation. However, because HOTAIR and YBX1 are both multifunctional molecules and other regulation pathways are sure to exist. However, the model proposed above can explain at least some of the effects of HOTAIR on cancer cells.

## Discussion

To date, many studies have reported the role of lncRNA HOTAIR as a regulator of cellular proliferation, apoptosis, and oncogenesis (Wan

& Chang, 2010; Zhang et al, 2014). The wide range of functions regulated by HOTAIR is attributable to its ability to interact with a variety of different proteins, but the exact mechanisms governing these functions remain to be fully characterized. The cellular function of an lncRNA is closely associated with its cellular localization. Nuclear lncRNAs mainly regulate the transcription or mRNA processing (Zhang et al, 2012; McHugh et al, 2015; Katsel et al, 2019), whereas cytoplasmic lncRNAs may exert cytosolic functions related to protein localization, post-transcriptional regulation, translation, and RNA stability (Faghihi et al, 2008; Sharma et al, 2011; Yoon et al, 2012; Guttman et al, 2013; Du et al, 2016). Some lncRNAs can also shuttle between different compartments in response to different cellular conditions (Derrien et al, 2012; Dong et al, 2017; Williams et al, 2018). HOTAIR is distributed both in cytoplasm and nuclear (Khalil et al, 2009; Xue et al, 2016), which was also validated by our RNA FISH results (Figs 2E and 4C and E). To better understand how HOTAIR influences cell biology, we conducted ChIRP-MS assays in entire cell lysates to fully screen the potential HOTAIR-interacting proteins. Through this approach we identified 348 putative HOTAIR-binding proteins, thereby greatly expanding current knowledge of HOTAIR-interacting proteins. We have compared our ChIRP-MS data with two in vivo RNA interactome capture studies which were also performed in HeLa cells (Castello et al, 2012; Trendel et al, 2019). Our dataset has 71 and 94 proteins overlapping with these two datasets, respectively (Table S1). However, several reported HOTAIR interacting proteins have not been identified, such as components in PRC2 complex (SUZ12, EED, and EZH2), and LSD1 (Gupta et al, 2010; Tsai et al, 2010). AKT and RSK which were found to interact with HOTAIR in the present study were also not detected by ChIRP-MS approach here. We speculated the main reason is that ChIRP-MS method tends to identify proteins with relatively high abundance. We retrieved the abundance percentile (AP) of proteins in ours and two other ChIRP-MS datasets, Xist ChIRP-MS (Chu et al, 2015) and RV RNA ChIRP-MS (Ooi et al, 2019), both performed in clonal derivatives of HeLa cells (HeLa S3 and H1HeLa, respectively), from the database of proteins abundance information from HeLa cells (Itzhak et al, 2016). The proteins from three ChIPR-MS data showed very similar AP distributions (Fig S8), the AP of 75–80% proteins in these datasets are larger than 80. The number of ChIRP-MS identified proteins with an AP < 80 is sharply decreased, demonstrating that proteins with a high AP (>80) are much more likely to be successfully screened in a ChIRP-MS experiment. The AP of AKT, RSK, LSD1, EZH2, and EED is 44.3, 77.1 71.8, 58.7, and 72.9, respectively. Because ChIRP-MS was performed in whole cell lysis, the signal of nuclear proteins such as LSD1 and EZH2 may be covered by high abundance cytoplasmic proteins. Moreover, different detecting approach may be another reason for not detecting previously reported HOTAIR-interacting partners (Tsai et al, 2010; Wu et al, 2013). Future experiments aimed at validating these interactions and exploring their implications for cell functionality will serve to better elucidate the myriad roles played by HOTAIR in regulating cell biology and cancer development.

Visualizing interactions between lncRNAs and proteins in live cells is essential for the accurate understanding of lncRNA functionality. We have previously developed two TriFC systems for in vivo imaging of RNA–protein interactions (Yin et al, 2013; Han et al, 2014). However, for lncRNAs like HOTAIR, which is more than 2,000 nt in length with a complex secondary structure essential for its

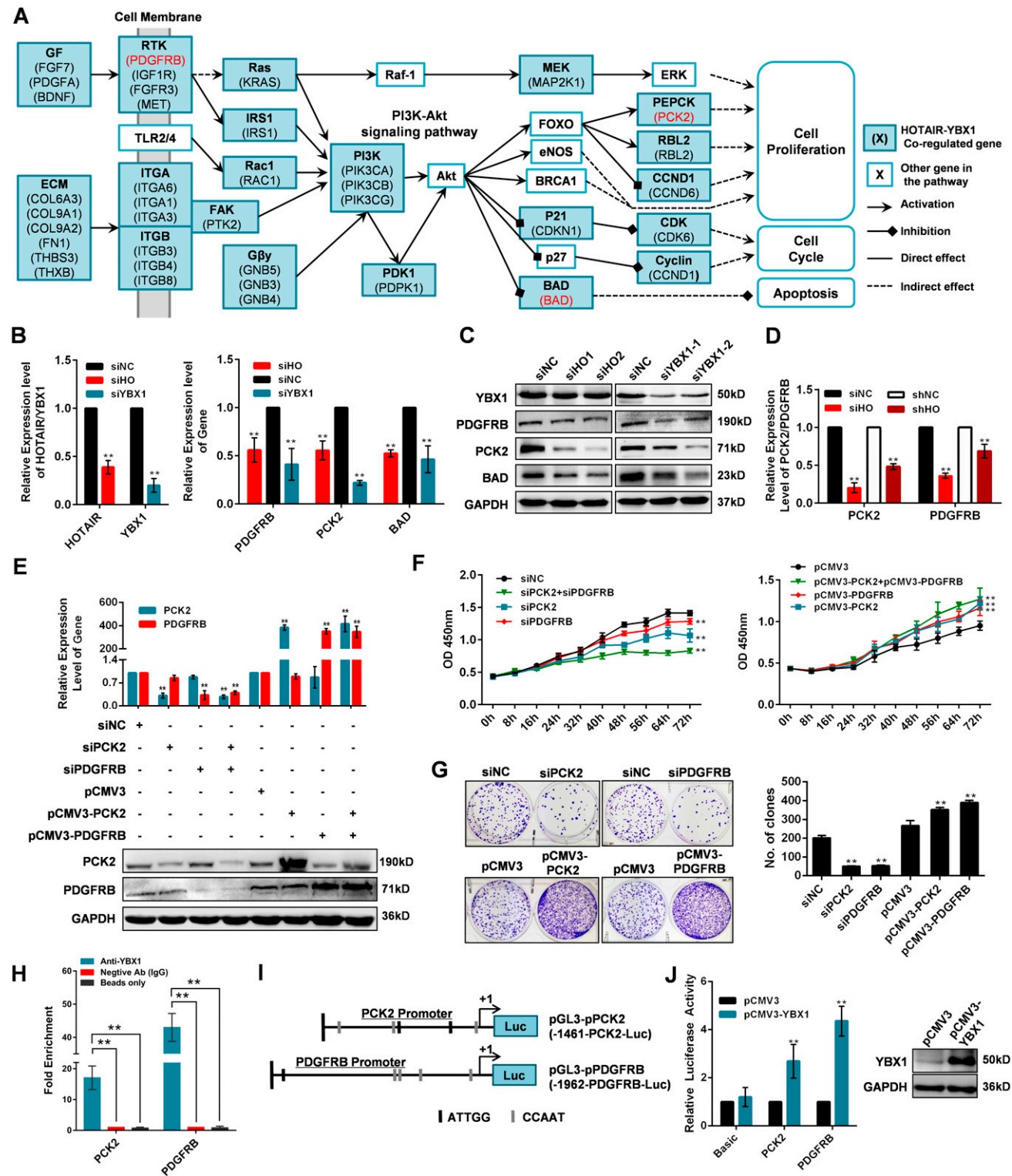

**Figure 5. HOTAIR and YBX1 co-regulate downstream gene transcription.**

**(A)** Putative HOTAIR-YBX1 co-regulated genes linked to the PI3K/Akt pathway. Genes indicated in red are those selected for further validation. **(B, C)** The RNA and protein expression change of platelet derived growth factor receptor β (PDGFRB), PCK2 and BAD in HOTAIR or YBX1 knocked down cells HeLa cells. **(D)** The mRNA expression of PCK2 and PDGFRB in HeLa tumor xenografts. HOTAIR transient knockdown HeLa cells (siHO group) or HOTAIR stable knockdown HeLa cells (shHO group) were injected subcutaneously near the axillary fossa of nude mice. HeLa cells transfected with siNC (siNC group), or shNC expressed HeLa cells (shNC) were also injected as negative controls. The mice were euthanized after 16 d, and the tumors were dissected. The mRNA expression of PCK2 and PDGFRB in tumors was measured by qRT-PCR. **(E)** Measurement of PCK2 and PDGFRB in HeLa cells 48 h after transfection using siPCK2/siPDGFRB/siNC, empty vector (pCMV3), plasmid-PCK2 (pCMV3-PCK2), or plasmid

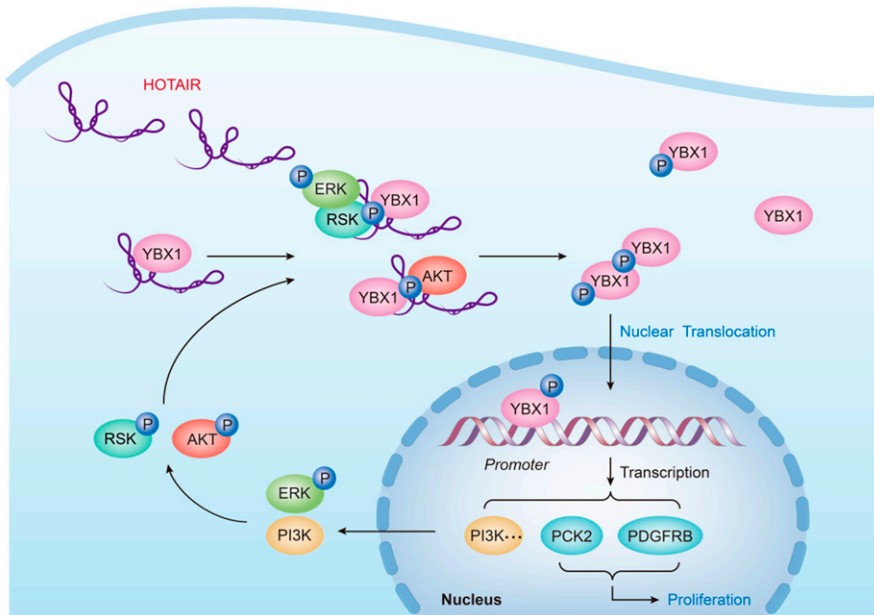

**Figure 6. A proposed model showing the functional crosstalk between HOTAIR and YBX1.**
Overexpression of HOTAIR leads to its interaction with both YBX1 and the kinases pAKT and pRSK, leading to enhanced YBX1 S102 phosphorylation. This in turn leads to YBX1 nuclear translocation, activating gene transcription, resulting in PI3K/ERK activation that drives further AKT and RSK phosphorylation in a positive feedback loop. This activated PI3K/Akt and ERK/RSK pathways further drive the phosphorylation and nuclear translocation of YBX1. HOTAIR is ultimately able to modulate cell proliferation in part via regulating the YBX1 target genes PCK2 and platelet derived growth factor receptor β.

functionality (Somarowthu et al, 2015), the extant TriFC systems are ill-suited because of the relatively high background, short wavelengths, and requirement for low temperature maturation (Yin et al, 2013; Han et al, 2014). We therefore chose to develop a novel iRFP-TriFC system. This system has longer wavelengths, lower background, greater brightness, and enabling us to visualize lncRNA-protein interactions under physiological temperature. The novel iRFP-TriFC system successfully validated the binding between HOTAIR and three potentially HOTAIR-interacting proteins identified by ChIRP-MS. We also used this system to validate the previously reported HOTAIR-interacting proteins, LSD1 and EZH2 (Gupta et al, 2010; Tsai et al, 2010). The results demonstrated that this iRFP-TriFC system represents a powerful experimental tool for visualizing and understanding lncRNA biology.

Among the hundreds of potentially HOTAIR-interacting proteins, YBX1 has been associated with a range of oncogenic processes including cell proliferation, tumor progression, genomic instability, metastasis, and drug resistance (Lasham et al, 2013). Moreover, YBX1 has also been reported to interact with multiple distinct oncogenic ncRNAs including lncRNAs (Castello et al, 2012; Diaz-Lagares et al, 2016; Li et al, 2017; Zhang et al, 2018), thereby regulating cancer progression and oncogene expression. YBX1 is thus thought to be a master regulator of RNA regulated cancer cell biology (Lyabin et al, 2014). In this study, we were able to validate the in vivo interaction between HOTAIR and YBX1 using the iRFP-TriFC system, RIP, and ChIRP approaches (Fig 2). Using our iRFP-TriFC system, we further

found that HOTAIR specifically interacts with the YBX1 CSD region via CACC motifs. To date YBX1 has been found to interact with multiple distinct oncogenic lncRNAs, including TP53TG1 (Diaz-Lagares et al, 2016), HULC (Li et al, 2017), and HOXC-AS3 (Zhang et al, 2018), thereby regulating cancer progression and oncogene expression. The specific mechanisms by which these lncRNAs regulate YBX1 transcriptional activity, however, remain poorly characterized. In this study we were able to show a specific interaction between HOTAIR and YBX1, and we demonstrated that this interaction mediates at least in part the known effects of HOTAIR on cancer cell biology.

The trafficking of YBX1 between the nucleus and the cytosol is closely linked to the transcription of YBX1 target genes in cancer cells (Eliseeva et al, 2011). Phosphorylation of the S102 residue in the CSD region of YBX1 is associated with its nuclear translocation (Sutherland et al, 2005). In addition, YBX1 has been shown to interact with the lncRNA TP53TG1, leading to the association of YBX1 with chromatin, thus indicating that lncRNAs can regulate YBX1 localization within cells (Diaz-Lagares et al, 2016). The lncRNA HULC (highly up-regulated in liver cancer) has also been shown to drive ERK-mediated YBX1 phosphorylation, thereby increasing the transcription of oncogenic genes in HULC-expressing cells (Li et al, 2017). In this study, we determined that HOTAIR promotes pAKT- and pRSK-mediated YBX1 S102 phosphorylation and subsequent nuclear translocation (Fig 4). Interestingly, we also determined that HOTAIR can interact with pAKT and pRSK in the cytoplasm (Fig 4).

PDGFRB (pCMV3-PDGFRB) as confirmed by qRT-PCR and Western blotting. **(F, G)** CCK-8 and colony formation assays in PCK2/PDGFRB knockdown/overexpressed HeLa cells. **(H)** ChIP assay followed by qPCR was used to detect the binding of YBX1 to the PCK2 and PDGFRB promoters. Nonspecific IgG and bead-only controls were used as negative controls. ChIP-qPCR data were normalized to fold enrichment method with IgG group used as background signal. **(I)** Schematic diagram of inserting PCK2 and PDGFR promoters into pGL3-Basic vector, with putative YBX1-binding sites (Y-boxes), ATTGG (black box), or CCAAT (gray box) indicated. **(J)** Luciferase reporter analysis of YBX1 binding to the *PCK2* and *PDGFRB* promoter regions. Western blotting was used to confirm YBX1 overexpression. **(E, G)** In (E, G), representative results were shown. **(B, D, E, F)** In (B, D, E, F), data are shown as means ± SD of triplicate experiments. *P < 0.05 and **P < 0.01 (t test).

Given these results, we proposed a model illustrating the molecular mechanisms governing HOTAIR biology (Fig 6). In cancer cells, the increased expression of HOTAIR allows this lncRNA to serve as a scaffold for the interaction of YBX1-pAKT and/or YBX1-pERK/pRSK, thereby leading to an increase in YBX1 S102 phosphorylation and subsequent nuclear translocation. This nuclear translocation of YBX1 leads to the increased transcription of YBX1 target genes, including those promoting cell growth and tumorigenesis. YBX1 is known to bind the PI3K promoter region, promoting PI3K transcription and activating the PI3K/Akt (Mayo et al, 2002; Sutherland et al, 2005; Aksamitiene et al, 2012). Consistent with these findings, we observed increased PI3K activation upon HOTAIR overexpression and YBX1 nuclear localization, leading to increased phosphorylation of the downstream target of PI3K and AKT. YBX1 binds to the promoters of a number of genes involved in the ERK/MAPK pathway (Imada et al, 2013; Donaubauer & Hunzicker-Dunn, 2016), and significantly alters expression of transcripts encoding MAPK phosphatases, such as MKP2 (Lasham et al, 2012). Interestingly the ERK/MAPK pathway and PI3K/Akt pathway can in turn lead to additional YBX1 phosphorylation in a positive feedback loop. In addition to PI3K, we found that HOTAIR-mediated YBX1 nuclear localization also drives the expression of the growth-promoting *PCK2* and *PDGFRB* genes. We confirmed a direct association between YBX1 and the *PCK2* and *PDGFRB* promoter regions by both ChIP and Luciferase reporter gene assays (Fig 5). Expression of these genes was also positively correlated with HOTAIR expression in both HeLa cells and in a xenograft mouse model (Figs 5B and D and S7). PCK2 is a mitochondrial isoform of phosphoenolpyruvate carboxykinase (PEPCK) and is known to be overexpressed in multiple human cancers (Montal et al, 2015). It has been found to promote both PCK2 cell growth and resistance to metabolic stress via metabolic reprogramming (Vincent et al, 2015). PDGFRB is thought to play a role in driving the growth and aggressiveness of multiple tumor types (Melaiu et al, 2017). By assessing how PCK2 and PDGFRB knockdown/overexpression influenced the growth of HeLa cells, we were able to demonstrate that HOTAIR-mediated effects on cell proliferation can be mimicked by modulating PCK2 and PDGFRB expression, suggesting that PCK2 and PDGFRB are responsible for regulating cell proliferation for a portion of the HOTAIR interactome.

Through the mechanisms described above, HOTAIR has the potential to regulate the expression of hundreds of genes, with YBX1 representing one key mechanism whereby HOTAIR can regulate cancer cell proliferation and signaling. Although we have uncovered interesting and novel biology in this study, the interactions of HOTAIR and YBX1 are very complex, with the specific interaction partners involved in these interactions having the potential to alter the resultant biological regulatory processes. As such, it is possible that the interactions between HOTAIR and YBX1 demonstrated in the present study may have broader context-dependent implications for cell biology, and such a possibility warrants further experimental investigation.

In conclusion, in this study, we have used a proteomic analysis to identify proteins in cancer cell which interact with the lncRNA HOTAIR, confirming the value of ChIRP-MS as a means for the functional characterization of the interactome of a given lncRNA. We further developed and optimized a novel iRFP-TriFC system suitable for the in vivo imaging and functional analysis of lncRNA–protein interactions. Using this system and complementary experimental techniques, we additionally found that HOTAIR regulates cancer cell proliferation at least partially via interacting with YBX1, altering its subcellular localization and target gene transcription. These results emphasize the importance of further exploring the HOTAIR protein interactome in cancer cells. Our mechanistic characterization of HOTAIR and its functional crosstalk with the well-established oncogene YBX1 may additionally serve to facilitate the development of cancer therapies targeting YBX1 through its interaction with HOTAIR.

# Materials and Methods

### Cell lines and mice

All cell lines used in this study were purchased from Shanghai Cell Bank, Chinese Academy of Sciences (Shanghai, China). Cells were grown in a standard incubator at 37°C and 5% $CO_2$ in Dulbecco's modified Eagle's high glucose medium (DMEM; HyClone) containing 10% FBS (Gibco). For animal studies, 5-wk-old, specific pathogen–free female BALB/c nude mice were purchased from Animal Center at Wuhan University (China) and were housed in a specific pathogen–free animal facility.

### Ethics statement

All animal work was conducted in a manner consistent with the guidelines of the Institutional Animal Care and Use Committee at Wuhan University.

### Cell transfection and treatment

Lipofectamine 2000 (Invitrogen) was used for all transfection reactions based on provided protocols. All siRNAs were designed and synthesized by Shanghai GenePharma Inc. Human HOTAIR mammalian expression vector pcDNA3.0-HOTAIR and the negative control pcDNA3.0-Antisense vector were generous gifts of Tsai et al (2010), Antisense refers to an antisense fragment of HOTAIR, which was designed and used by Tsai et al before (Tsai et al, 2010). The human *YBX1*, *PCK2*, and *PDGFRB* ORF mammalian expression plasmids (pCMV3-YBX1, pCMV3-PCK2, and pCMV3-PDGFRB) were purchased from Sino Biological Inc., with pCMV3-C-FLAG (pCMV3) as a negative control. For all transfection reactions, we confirmed the efficiency of knockdown or overexpression via qRT-PCR and Western blotting as appropriate. Two siRNAs were designed for transient transfection of HOTAIR (siHO1 and siHO2), YBX1 (siYBX1-1 and siYBX1-2), PCK2 (siPCK2-1 and siPCK2-2), and PDGFRB (siPDGFRB-1 and siPDGFRB-2), respectively. siHO indicates a 1:1 mixture of siHO1 and siHO2. siYBX1 indicates a 1:1 mixture of siYBX1-1 and siYBX1-2. siNC indicates negative control siRNA. Stable HOTAIR knockdown was achieved as previously described (Zheng et al, 2015), and two stably transfected cells (shHO1 and shHO2) were maintained in complete media supplemented with 1,000 ng/ml G418 disulfate. shHO indicates a 1:1 mixture of shHO1 and shHO2

knockdown cells. shNC indicates HeLa cells stably transfected with negative control shRNA. All siRNA and primer sequences are listed in Table S6.

To transfect cells with iRFP-TriFC plasmids, HeLa cells were seeded into glass-bottom dishes at ~75% confluency. A total of 1 $\mu$g of each component was used for the transfection reaction, and after transfection, the cells were allowed to rest for 18–20 h before imaging. For imaging, cells were administered 25 $\mu$M Biliverdine (BV; Toronto Research Chemicals) for 2 h before image capture with a Leica SP8 microscope.

For stimuli and inhibitor treatment, cells were treated with serum starvation for 24 h and then stimulated with 100 ng/ml EGF or IGF-I for 2 h. Cells were also treated with 100 $\mu$M LY294002, 5 $\mu$M Akt/protein kinase B signaling inhibitor 2 (API-2), 50 $\mu$M PD098059, or 25 $\mu$M SL0101 for 1 h before stimulation with IGF-I or EGF.

### Real-time quantitative reverse transcription-PCR (qRT-PCR)

The EZNA Total RNA Kit I (Omega) was used to extract total cellular RNA. We then synthesized first-strand cDNA with the PrimeScript RT Reagent Kit (Takara). qRT-PCR was then conducted using a Light-Cycler 480 Real-Time PCR system (Roche) and SYBR Green PCR Master Mix (Roche). GAPDH served as a normalization control.

### Cell proliferation assays

Cells were seeded at $2 \times 10^2$ cells/well in a 96-well plates, and cell viability was then measured via CCK-8 assay (Yeasen) after 0, 8, 16, 24, 32, 40, 48, 56, 64, and 72 h. Six samples were measured per condition. In colony formation assays, cells were also seeded at $2 \times 10^2$ cells/well, and cell numbers were determined after 21 d with the ImageJ software (National Institutes of Health).

### Xenograft mouse model and tumor processing

A total of 32 nude BALB/c mice were randomized into four groups (siHO, siNC, shHO, or shNC) and were then injected subcutaneously near the axillary fossa with $5 \times 10^6$ transient or stably transfected HeLa cells. After 16 d, the mice were euthanized by cervical dislocation, and then tumors were collected and stored at –80°C for future RNA and protein analysis.

### Wound healing assay

Cells plated in 35-mm dishes were allowed to achieve 90% confluency, at which time a sterile 200 $\mu$l pipette tip was used to generate a scratch wound in the cell monolayer. Cells were then grown in DMEM supplemented with 2% FBS, and an inverted microscope was used to image the cells at 0 and 72 h post-wound generation. The ImageJ software package was then used to quantify wound healing based on the change in wound area over time.

### Cell invasion assay

Matrigel (BD Biosciences) was used to coat chambers with an 8.0-$\mu$m pore size (Corning) in a 24-well plate. After 2 h incubation at 37°C, we added a total of $3 \times 10^5$ cells in serum-free medium to the upper chamber. The lower chamber was filled with complete media to serve as a chemoattractant, and cells which migrated to this chamber were identified via crystal violet staining. After staining, cells in five random microscopic fields were counted via inverted microscope.

### Cell apoptosis assay

After growing in six-well plates under appropriate experimental conditions, cells were collected and then stained using an Annexin V, FITC Apoptosis Detection Kit (Dojindo). After staining, a BD FACScan flow cytometer was used for sample analysis.

### Comprehensive identification of RNA-binding proteins (ChIRP)

ChIRP experiment was performed as previously reported (Chu et al, 2015; Chu & Chang, 2018). Fig 1A is an overview of the ChIRP strategy. Briefly, 20-mer DNA probes for HOTAIR were designed with single-molecule FISH online designer (https://www.biosearchtech.com/products/rna-fish/chirp-probe-sets) (Table S6). Cells were harvested and crosslinked by 3% formaldehyde. About 20 million cells were used for one ChIRP experiment. Cells were dissolved in cell lysis buffer (RIPA lysis buffer, Medium, Beyotime, add 1 mM PMSF (Thermo Fisher Scientific), protease inhibitor cocktail (Thermo Fisher Scientific), and SUPERase.In RNase Inhibitor (Thermo Fisher Scientific) fresh before use, 100 mg cell pellet/1 ml lysis buffer) and incubated on rotator at 4°C for 20 min, after which sonicate was used to break DNA in cell lysate to 1–2 kb fragments. High speed centrifugation was used to remove cellular debris. Each sample was precleared with 30 $\mu$l washed streptavidin C1 beads (Invitrogen) on rotator at 37°C for 30 min. For RNase treatment control, we added RNase A to the lysate (10 $\mu$g/ml final concentration). Then 2 × volume hybridization buffer and ChIRP probes were added to cleared samples and incubated at 37°C with end to end rotation for 5 h. Two sets of negative control groups were used for ChIRP-MS experiment, RNase A treatment and nontargeting probe control. Nontargeting probe control refers to using a nontargeting probe that does not bind any human RNA (Kaida et al, 2010; Berg et al, 2012) (Table S6). When validating the interaction between HOTAIR and a specific protein using ChIPR followed by Western blotting, in addition to the two negative controls above, another negative control group using probe targeting an unrelated RNA U2 was also used. C1 beads were washed three times on a magnet stand with cell lysis buffer (use 100 $\mu$l beads per 1 $\mu$l of 100 $\mu$M probes) before completion of hybridization. Then 1 ml of sample was used to resuspend and transfer beads into the reaction and then continue to mix samples at 37°C for 30 min. The samples were washed five times with 500 $\mu$l wash buffer (2× SSC, 0.05% SDS, add 1 mM PMSF fresh before use. vol/vol to original beads volume). Proteins are eluted twice from beads in 100 $\mu$l biotin elution buffer (12.5 mM D-biotin, 7.5 mM Hepes (pH 7.5), 75 mM NaCl, 1.5 mM EDTA, 0.015% SDS, 0.075% sarkosyl, and 0.02% Na-Deoxycholate.) at room temperature with mixing for 20 min, and then at 65°C for 10 min. The proteins were precipitated in TCA overnight at 4°C. The next day, we pelleted proteins and washed protein pellets once with cold acetone. 1× Laemmli sample buffer was added to protein pellets and boiled for 30 min to reverse the cross-links. Final protein samples

were separated in SDS–PAGE gels for silver staining, mass-spectrometry or Western blotting.

## Protein separation

For LC–MS/MS analysis, proteins isolated via CHIRP were run on an SDS–PAGE gel and silver stained. Each lane of the gel was then cut into five sections and subdivided into cubes of 1 mm$^3$ each. These samples were then subjected to in-gel digestion as discussed previously (Zheng et al, 2015). A SpeedVac was then used to dry the combined peptide extracts from these gel segments.

## Nano liquid chromatography (LC)-tandem mass spectrometry (MS/MS)

An Easy-nLC chromatography system (Thermo Fisher Scientific) was used for peptide separation. The peptide mixture was separated with a linear gradient of 5–60% buffer B (80% ACN and 0.1% formic acid) over 90 min at a flow rate of 250 nl/min on a C$_{18}$-reversed phase column (75 $\mu$m ID × 10 cm) packed in-house with ReproSil-Pur C$_{18}$-AQ 3 m resin (Dr. Maisch GmbH) in buffer A (0.1% acetic acid). This setup was extended with a trap column (100 $\mu$m ID × 2 cm) containing identical chromatographic material. The Easy-nLC chromatography system was on-line coupled to the Orbitrap Elite instrument (Thermo Fisher Scientific) via a Nanospray Flex Ion Source (Thermo Fisher Scientific) in positive ion mode at the spray voltage of 2.2 kV. The heated capillary was maintained at a temperature of 320°C. MS data were acquired in a data-dependent strategy selecting the fragmentation events based on the precursor abundance in the survey scan (300–1,300 D). The resolution of the survey scan was 60,000 at $m/z$ 400 D with a target value of 1 × 10$^6$ ions and one or two microscans. HCD MS/MS spectra were acquired with a resolution of 15,000, with a target value of 1 × 10$^5$ ions or a maximum integration time of 100 ms, setting the first mass to 120 Da. Dynamic exclusion was 30 s and early expiration was disabled. For the MS/MS fragmentation, 28% of normalized collision energy and 2 D of isolation width were used.

## MS/MS data analysis

The analysis of the mass spectrometric was carried out using The MaxQuant Software v1.5.8.3 (Thermo Fisher Scientific). The UniProt human protein database (UP000005640, 74349 sequences, release 2019_07) with decoy sequences was used. Database search parameters were: Fixed modifications, Carbamidomethyl (C), Variable modifications, Oxidation (M), Acetyl (Protein N-term), and Deamidated (NQ); precursor ion mass tolerance, 6 ppm after refinement by Maxquant; MS/MS mass tolerance, 0.5 D; Enzyme specificity, trypsin/p; maximum missed cleavages, 2. Protein identification criteria were: (1) peptide length ≥ 6, (2) FDR ≤ 1% at both the PSM and protein levels. Unique peptides ≥ 2. To identify high-confidence proteins which interact with HOTAIR, datasets from three biological replicates and three negative controls were analyzed with SAIN-Texpress (v3.6.1) (Teo et al, 2014). Those prey proteins which included in both three biological replicates but not in any of negative replicate were determined to be high confidence hits.

## GO enrichment and protein–protein interaction analyses

The plug-in of Cytoscape v3.6.1, Bingo, was used for GO enrichment analysis. An analysis of PPIs was conducted with the STRING database v10.0 (Szklarczyk et al, 2015), whereas Cytoscape v3.6.1. (Shannon et al, 2003) was used to visualize results. Function analyses of proteins involved in STRING networks were performed using DAVID (https://david.ncifcrf.gov/).

## Protein extraction

Moderate Western and IP lysis buffer (Beyotime) containing PMSF (Beyotime) and protease inhibitor cocktail (Thermo Fisher Scientific) was used for isolation of total cellular protein. Samples were allowed to lyse for 30 min at 4°C on a rotator, after which high speed centrifugation was used to remove cellular debris. For the collection of specific subcellular fractions, the NE-PER Nuclear and Cytoplasmic Extraction Reagents kit (Thermo Fisher Scientific) was used based on the provided instructions. A BCA Protein Assay Kit (TANGEN) was used to quantify all protein samples.

## Western blotting

Protein was detected using primary antibodies detecting YBX1, CDK6, TCTP, GAPDH, H3, and pYBX1 (ab138654 and ab138654 detect endogenous levels of YB1 only when phosphorylated at S102), pAKT, pERK, pRSK, AKT, ERK, RSK, PDGFRB, PCK2, and BAD. Anti-pAKT, anti-pERK, anti-pRSK, anti-AKT, anti-ERK, and anti-RSK were purchased from Cell Signaling Technology (CST) and all the other antibodies were purchased from Abcam. Western blotting was performed as in previous studies (Li et al, 2016). Briefly, protein samples collected above were separated on SDS–PAGE gels via electrophoresis, transferred to PVDF membranes (Millipore), blocked, and probed using appropriate primary antibodies. Samples were then probed with appropriate secondary antibodies and protein bands were detected using the ECL reagent (GE Healthcare). GAPDH was used as a loading control for total and cytoplasmic protein, whereas H3 was used for nuclear protein. ImageJ was used to quantify protein band densitometry.

## RNA immunoprecipitation (RIP)

Immunoprecipitation of YBX1/CDK6/TCTP/AKT/ERK/RSK was performed as previously described with slight modification (Selth et al, 2009). Briefly, HeLa cells were treated with formaldehyde to cross-link in vivo protein–RNA complexes for 5 min, and glycine was added to stop the cross-linking. Cells were washed and lysed by sonication in RIP buffer (150 mM KCl, 25 mM Tris, pH 7.4, 5 mM EDTA, 0.5 mM DTT, 0.5% NP-40, and 100 U/ml RNAase inhibitor). An aliquot of lysate was frozen for input RNA isolation. Cell lysates was precleared with 15 $\mu$l Dynabeads Protein G beads (Invitrogen), and Dynabeads Protein G beads were precoated with anti-YBX1, anti-CDK6, anti-TCTP, anti-AKT, anti-ERK, anti-RSK, or negative control rabbit IgG antibody. Then the precleared lysate was incubated with precoated beads for 4 h at 4°C. All the antibodies were purchased from Abcam. After precipitation, beads were collected and washed with four times with washing buffer I (50 mM Tris, pH 7.4, 1 M NaCl, 1%

NP-40, and 1% sodium deoxycholate), and four times with washing buffer II (50 mM Tris, pH 7.4, 1 M NaCl, 1% NP-40, 1% sodium deoxycholate, and 1 M Urea). Coprecipitated RNAs was isolated by resuspending beads in 1 ml TRIzol (Thermo Fisher Scientific) according to the manufacturer's instruction. RNA was purified by phenol/isopropanol precipitation, followed by DNase treatment. RNA was then quantified, and equal amounts were analyzed via qRT-PCR.

## Plasmid construction

To develop the iRFP-TriFC system, iRFP was split either between amino acids 97/98 or 123/124, yielding the iRN97 (aa 1–97), iRC98 (aa 98–316), iRN123 (aa 1–123), and iRC124 (aa 124–316) fragment pairs. Fusion protein expression plasmids were constructed based on the mNeptune TriFC system (Han et al, 2014). The iRN97/iRN123 and iRC98/iRC124 coding regions were PCR-amplified from the pcDNA3.1-iRFP plasmid. The pNS1-iRN97/iRN123 or pPTB-iRN97/iRN123 plasmids were then produced via replacing the mNeptune fragments MN155 in pNS1-155MN or pPTB-155MN with iRN97/iRN123. The iRC fragment was fused to the bacteriophage MCP. The piRC98/piRC124-MCP plasmid was generated by replacing the mNeptune fragment MC156 in pMC156-MCP with iRC98/iRC124. The pECFP-ms2, pECFP-ms2-M 5′UTR, pECFP-ms2-NP 5′UTR, and pECFP-ms2-env CRS constructs were identical to those used with the mNeptune-TriFC system (Han et al, 2014), whereas pEGFP-iRN123 was the same as that used for the iRFP-BiFC system (Chen et al, 2015).

To construct piRC124m-EGFP and piRC124m-MCP, the iRFP C-terminal coding region underwent site-directed mutagenesis at F165Y (TTC-TAC), D232Y (GAC-TAC), and W309R (TGG-CGG) with specific PCR primers. This iRC124m fragment was inserted into pEGFP-N1 to yield iRC124m-EGFP. The piRC124m-MCP plasmid was constructed by exchanging the iRC124 fragment in piRC124-MCP for iRC124m.

pYBX1-iRN123, pCDK6-iRN123, pTCTP-iRN123, pLSD1-iRN123, and pEZH2-iRN123 were constructed by replacing the PTB coding sequence in pPTB-iRN123 with the coding sequences of YBX1, CDK6, TCTP, LSD1, and EZH2, respectively.

To determine the interaction domain of YBX1 with HOTAIR, we constructed three YBX1 deletion mutants representing three primary domains of YBX1—A/PD (1–51 aa), CSD (52–129 aa), and CTD (130–324 aa) (Goldstein et al, 1990). pA/PD-iRN123, pCSD-iRN123, and pCTD-iRN123 were constructed by replacing the YBX1 coding region in pYBX1-iRN123 with the A/PD, CSD, or CTD coding sequence, respectively.

For the construction of pECFP-ms2-HOTAIR, pECFP-ms2-Antisense and pECFP-ms2-G3BP1, the coding sequence of HOTAIR/Antisense/G3BP1 was PCR-amplified from pcDNA3.0-HOTAIR, pcDNA3.0-Antisense, or HeLa cDNA, and then inserted into the pECFP-ms2 plasmid.

To determine which HOTAIR domains were important for YBX1 interaction, each of the four HOTAIR domains (HD1: 1–530 nt, HD2: 531–1,040 nt, HD3: 1,041–1,513 nt, HD4:1,514–2,149 nt [Somarowthu et al, 2015]) was PCR-amplified and cloned into the pEGFP-ms2 plasmid. Two motifs CACC and CAUC were reported to be high-affinity YBX1 binding sites (Wei et al, 2012; Ray et al, 2013). There are 13 CACC motifs (4 in HD1, five in HD2 and four in HD4) and three CAUC

motifs (2 in HD2, and one in HD4) in HOTAIR. We generated point mutations in the second A and the fourth C of CACC or CAUC motifs (CACC > CTCG, CAUC > CUUG) in HOTAIR. The three CACC mutated fragments HD1m, HD2m, and HD4m were generated by gene synthesis. A full-length HOTAIR containing all of these mutated sites (HOTAIRm) was then generated by fusion PCR using HD1m, HD2m, HD3 and HD4m. HOTAIRm′ fragments were generated by site-directed mutagenesis at CAUC sites of HOTAIR with specific PCR primers. pECFP-ms2-HD1, pECFP-ms2-HD2, pECFP-ms2-HD3, pECFP-ms2-HD4, pECFP-ms2-HD1m, pECFP-ms2-HD2m, pECFP-ms2-HD4m, pECFP-ms2-HOTAIRm, and pECFP-ms2-HOTAIRm′ were constructed by inserting HD1, HD2, HD3, HD4, HD1m, HD2m, HD4m, HOTAIRm, and HOTAIRm′ fragments into the pECFP-ms2 plasmid.

To construct pECFP-ms2-HD234, pECFP-ms2-HD34, HOTAIR HD234 (the truncated HOTAIR including HD2, HD3, and HD4 domains), and HD34 (the truncated HOTAIR including HD3 and HD4 domains) were amplified by PCR from pcDNA3.0-HOTAIR plasmids and then inserted into the pECFP-ms2 plasmid.

To construct pcDNA3.0-HOTAIRm, pcDNA3.0-HD234, and pcDNA3.0-HD34, HOTAIRm, HD234, or HD34 were inserted into the pcDNA3.0 plasmid.

To construct pEGFP-YBX1, pEGFP-YBX1 S102A, and pEGFP-YBX1 S102D, YBX1 was amplified by PCR from cDNA template, whereas YBX1 S102A or S102D mutations were achieved via site-directed mutagenesis. All coding genes were then inserted into the pEGFP-C1 plasmids.

To construct pAKT-iRN123, pRSK-iRN123, pERK-iRN123, pAKT D1-iRN123, pAKT D2-iRN123, pAKT D3-iRN123, pRSK D1-iRN123, pRSK D2-iRN123, pRSK D3-iRN123, AKT/RSK/ERK, and their domains were PCR-amplified from pCMV3-AKT, pCMV3-RSK, or pCMV3-ERK plasmids, and replacing the PTB coding sequence in pPTB-iRN123.

To verify the successful expression of recombinant protein PTB-iRN97, NS1-iRN97, iRC98-MCP, PTB-iRN123, NS1-iRN123, iRC124-MCP, iRC124m-MCP, YBX1-iRN123, A/PD-iRN123, CSD-iRN123, CTD-iRN123, AKT-iRN123, AKT-D1-iRN123, AKT-D2-iRN123, AKT-D3-iRN123, RSK-iRN123, RSK-D1-iRN123, RSK-D2-iRN123, RSK-D3-iRN123, and ERK-iRN123, the coding sequences of recombinant proteins were PCR-amplified and introduced to pCMV-Flag vector by homologous recombination.

The pGL3-Basic and pRL-TK plasmids, containing luciferase and Renilla luciferase reporters, respectively, were purchased from Promega. The promoter sequences (–1,461 to +48 bp of the human PCK2 promoter [pPCK2] and –1,962 to +48 bp of the human PDGFRB promoter [pPDGFRB]) were downloaded from the UCSC Genome Browser (http://genome.ucsc.edu/index.html). Total HeLa DNA was then isolated via the TIANamp Genomic DNA Kit (TIANGEN), and these promoters were amplified by PCR from the HeLa cell genome and inserted into the pGL3-Basic plasmid to construct the pGL3-pPCK2 (–1893-PCK2-Luc) and pGL3-pPDGFRB (–1962-PDGFRB-Luc) plasmids.

The GGGGSGGGGS linker sequence was used to connect coding regions for this iRFP-TriFC system. Primer sequences are given in Table S6. DNA sequencing was used to confirm all sequences.

## Fluorescence microscopy

A Leica SP8 Confocal microscope (Leica Microsystems) with a 63×, 1.4 NA, oil immersion objective lens was used for all cell imaging.

Excitation wavelengths for EGFP, ECFP, iRFP, and Hoechst 33342 were 488, 445, 640, and 405 nm, respectively. All TriFC fluorescent staining was conducted in live cells, and quantitative analysis of resultant fluorescence was performed as described previously (Chen et al, 2015). ECFP fluorescence was used as an internal control to measure the efficiency of TriFC. The fluorescence ratios of iRFP/ECFP were quantified for every cell expressing the corresponding proteins after subtraction of the background fluorescence. Background fluorescence was calculated as the mean intensity of a 50 × 50 pixel$^2$ area of the image with no cells. About 50 cells from 10 microscopic fields for each sample were evaluated to quantify the fluorescence intensity.

### Immunofluorescent staining and RNA in situ hybridization

Cells grown in glass-bottom dishes were washed and then ice-cold ethanol was used for fixation, followed by appropriate permeabilization and blocking steps. Cells were then probed using rabbit anti-YBX1 (CST) followed by Alexa Fluor 488–conjugated secondary goat anti-Rabbit IgG (H+L) (Invitrogen). Hoechst 33342 was used for nuclear staining.

RNA FISH was performed based on previous protocols using a HOTAIR-AF647 probe (Invitrogen; see Table S6 for sequences). Briefly, after fixation and permeabilization, cells were combined with denatured FISH probes in hybridization buffer (50% formamide in 2 × SSC). Cells were then washed and imaged as described above.

### Co-immunoprecipitation (Co-IP)

Co-IP conducted as described previously (Li et al, 2016). Briefly, the Dynabeads Protein G (Invitrogen) was incubated with anti-YBX1 (Abcam) or negative control IgG (Abcam) diluted in PBS for 4–6 h at 4°C on a rotator. Supernatants were then discarded, and the beads coated with antibody were incubated with cell lysates for 10–12 h at 4°C on a rotator. The beads were then washed three times, and coprecipitated proteins were eluted and denatured with loading buffer, and then boiled for 5 min for downstream Western blot analyses.

### AKT/RSK kinase assay

A purified YBX1 CSD domain was purchased from ProteinGene Biotech. Kinase assays for AKT/RSK and YBX1 CSD were performed using a luminometric kinase assay with ADP-Glo Kinase Assay Kit (Promega). The AKT/RSK kinase was purchased from Promega Corperation. HOTAIR RNA was added to the reaction to check whether it can enhance the Kinase reaction, with Antisense, HD234 and HD34 used as negative control. In brief, the RNA was denatured at 65°C for 5 min and cooled to room temperature in the presence of RNA structure buffer. AKT/RSK and YBX1 CSD at optimal concentrations were incubated with HOTAIR, Antisense, HD234, or HD34 of equal molar concentration, in the presence of ATP. The kinase assay was performed at room temperature for 60 min. Then ADP-Glo Reagent was added to the reaction, and incubated for 40 min at room temperature. Finally, the Kinase Detection Reagent was added and the reaction mixture was incubated for another 40 min at ambient temperature. Then luminescence was recorded on an Orion L Microplate Luminometer (Berthold Technologies).

### Chromatin immunoprecipitation (ChIP) assay

ChIP was conducted as previously described (Chen, 2011). Briefly, we cross-linked the proteins to DNA by adding formaldehyde, and then terminated the cross-linking reaction by adding Glycine. Then cells were washed were washed and harvested in PBS. Chromatin was extracted by resuspending cells in a ChIP lysis buffer followed by sonication. This isolated DNA was then combined with 5 μg anti-YBX1 (CST). Blocking was then performed using herring sperm DNA (Invitrogen) and BSA (1 ng/μl), after which Magna ChIP Protein A+G Magnetic Beads (Millipore) were used to immunoprecipitate DNA-antibody complexes. DNA was then eluted, cross-linking was reversed, and DNA was quantified for use in qPCR. Nonspecific IgG and bead-only controls were used as negative controls. ChIP-qPCR data were normalized to fold enrichment method with IgG group used as background signal. Primer sequences are given in Table S6.

### Dual luciferase reporter assay

Cells were plated in 24-well plates and cultured overnight, then co-transfection with pCMV3-YBX1/pCMV3 (1 μg/well) and pGL3-pPCK2/pGL3-pPDGFRB (0.5 μg/well). Co-transfection pCMV3 and pGL3-pPCK2/pGL3-pPDGFRB was used as negative control. Luciferase activity was measured with the Dual-Luciferase Reporter Assay System (Promega) based on provided protocols. Transfection efficiency was normalized via the co-transfection of pR-TK (0.05 μg/well).

### Statistical rationale

For all the experiments carried out in this study, including qRT-PCR, cell proliferation assay, cell cycle progression assay, wound healing assay, cell invasion assay, cell apoptosis assay, ChIRP-MS, Western blotting, RIP, iRFP-TriFC system fluorescence microscopy, Immunofluorescent staining and RNA FISH, Co-IP, kinase assay, ChIP assay, and Dual Luciferase Reporter Assay, at least three independent biological replicates were performed. For xenograft mouse model and tumor processing experiment, eight BALB/c mice were used in each group. Statistical analyses were performed using GraphPad Prism version 6.01 (GraphPad Software). t test (two-tailed) was used to calculate the significance of differences among different groups. All data are presented as the means ± SD, a P-value < 0.05 was considered to be statistically significant.

## Data Availability

The MS proteomics data have been deposited to the ProteomeXchange Consortium with the dataset identifier PXD015546.

## Supplementary Information

# Acknowledgements

This work was supported by the National Natural Science Foundation of China (grant No. 31870756 and 91859108), the Strategic Priority Research Program of the Chinese Academy of Sciences (No. XDB29050201). The authors would like to thank Ms Ming Wang for her help in proteomic experiments and the Analysis and Testing Center of Institute of Hydrobiology.

## Author Contributions

S Li: data curation, formal analysis, investigation, and methodology.
Q Xiong: data curation, formal analysis, methodology, and writing—review and editing.
M Chen: data curation and methodology.
B Wang: investigation and methodology.
X Yang: investigation and methodology.
M Yang: investigation.
Q Wang: resources.
Z Cui: conceptualization, funding acquisition, and writing—original draft.
F Ge: conceptualization, data curation, formal analysis, supervision, funding acquisition, investigation, project administration, and writing—original draft, review, and editing.

## Conflict of Interest Statement

The authors declare that they have no conflict of interest.

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
