## [Reviewer comments · Life Science Alliance]

Life Science Alliance

Long non-coding RNA HOTAIR interacts with Y-Box Protein-1 (YBX1) to regulate cell proliferation

Siting Li, Qian Xiong, Minghai Chen, Bing Wang, Xue Yang, Mingkun Yang, Qiang Wang, Zongqiang Cui, and Feng Ge

DOI: <https://doi.org/10.26508/lsa.202101139>

Corresponding author(s): Feng Ge, Chinese Academy of Sciences

Review Timeline:

Submission Date:	2021-06-18
Editorial Decision:	2021-06-23
Revision Received:	2021-06-29
Accepted:	2021-07-01

Transaction Report:

Please note that the manuscript was reviewed at Review Commons and these reports were taken into account in the decision-making process at Life Science Alliance.

Review
COMMONS

June 23, 2021

RE: Life Science Alliance Manuscript #LSA-2021-01139-T

Feng Ge
Chinese Academy of Sciences
Institute of Hydrobiology
Wuhan, 430072, China

Dear Dr. Ge,

Thank you for submitting your revised manuscript entitled "long non-coding RNA HOTAIR interacts with Y-Box Protein-1 (YBX1) to regulate cell proliferation". We would be happy to publish your paper in Life Science Alliance pending final revisions necessary to meet our formatting guidelines.

- please upload your main and supplementary figures as single files
- please add a Category and a Summary Blurb/Alternate Abstract for your manuscript in our system
- please label the Summary as "Abstract"
- please consult our manuscript preparation guidelines <https://www.life-science-alliance.org/manuscript-prep> and make sure your manuscript sections are in the correct order
- please add an Author Contributions section to your main manuscript text
- please add your main, supplementary figure, and table legends to the main manuscript text after the references section;
- there is a mention of panel F in the legend for Figure 3 (although there is no such panel in the actual figure). Please check
- please add callouts for Figures 5J and S4A-C to your main manuscript text
- please add a conflict of interest statement to your main manuscript text

Figure checks:

- missing scale bars for Figure S1J, S4B
- please add molecular weights next to all blots in the figures

A. FINAL FILES:

B. MANUSCRIPT ORGANIZATION AND FORMATTING:

Sincerely,

Eric Sawey, PhD
Executive Editor

July 1, 2021

RE: Life Science Alliance Manuscript #LSA-2021-01139-TR

Prof. Feng Ge
Chinese Academy of Sciences
Institute of Hydrobiology
Wuhan, 430072, China
Wuhan 430072
China

Dear Dr. Ge,

Thank you for submitting your Research Article entitled "Long non-coding RNA HOTAIR interacts with Y-Box Protein-1 (YBX1) to regulate cell proliferation". It is a pleasure to let you know that your manuscript is now accepted for publication in Life Science Alliance. Congratulations on this interesting work.

DISTRIBUTION OF MATERIALS:

Again, congratulations on a very nice paper. I hope you found the review process to be constructive and are pleased with how the manuscript was handled editorially. We look forward to future exciting submissions from your lab.

Sincerely,
